# Unpacking sources of transmission in HIV prevention trials with deep-sequence pathogen data

Lerato E. Magosi [1,2] ✉, Eric Tchetgen Tchetgen[3], Vlad Novitsky[4,5], Molly Pretorius Holme[4], Janet Moore[6], Pam Bachanas[6], Refeletswe Lebelonyane[7], Christophe Fraser[8], Sikhulile Moyo [5], Kathleen E. Hurwitz[9], Tendani Gaolathe[5], Ravi Goyal[10], Joseph Makhema[5], Shahin Lockman[4,5,11,14], Max Essex[4,5,14], Victor De Gruttola [12,14] & Marc Lipsitch [1,13,14] ✉

To develop effective HIV prevention strategies to guide public health policy the main sources of infection in HIV prevention studies must be identified. Accordingly, we devised a statistical approach that estimates the relative contribution of different sources of infection in community-randomized trials of infectious disease prevention using deep- (or next generation) sequenced pathogen data. We applied this approach to the Botswana Combination Prevention Project (BCPP) and estimated that 90% [95% Confidence Interval (CI): 80–94] of new infections in communities that received combination prevention (including universal HIV test-and-treat) originated from individuals residing in communities outside the trial area. We estimate from our model that the relative benefit of providing the BCPP intervention to all communities nationwide would be a 59% [3–87] reduction in transmissions to recipients in trial communities, exceeding the 30% reduction observed when providing the BCPP intervention to trial communities only. Our results suggest that the impact of the BCPP trial intervention was curtailed by sources of transmission outside the trial area and could be considerably larger if applied nationally. We recommend that the impact of sources of transmission beyond the reach of the intervention be considered when designing and evaluating interventions to inform public health programs.

Why did the landmark community-randomized universal HIV test-and-treat trials in sub-Saharan Africa - BCPP/ Ya Tsie[1], HPTN 071/ PopART [2], SEARCH[3] and ANRS 12249/TasP[4] - show variable reductions in the occurrence of new HIV infections in trial communities that received the intervention compared to control communities (0–30%) despite substantial gains in viral suppression [5]? This is one of the most important questions in the HIV policy world today because HIV "test-and-treat" was thought to hold great potential to bring the HIV epidemic under control in the absence of a successful vaccine or functional cure. Some of the variation in the incidence reductions observed is thought to be due to a change in national HIV treatment guidelines to universal treatment part-way through the trials effectively reducing the difference between intervention and control communities. Another complementary hypothesis is that HIV transmissions to residents of intervention communities from individuals in non-intervention communities in the trial (control communities) and from communities not taking part in the trial (non-trial communities) limited the size of effect observed in the trials, but it is unknown to what degree. A large dilution of the intervention effect in the trials by transmission from non-intervention communities

**Table 1 | Negative-binomial regression models describing the expected probability of viral linkage between a pair of individuals randomly sampled from their respective communities in the BCPP trial**

| Variable | Coefficient | Standard Error | 95% Conf. Interval | P value |
|---|---|---|---|---|
| Baseline model: Before the intervention had taken effect | | | | |
| Intercept | −11.59 | 0.54 | −12.65 to −10.53 | <0.001 |
| Transmission source: control community | 0.63 | 0.42 | −0.20 to 1.45 | 0.14 |
| Transmission type: same community | 3.56 | 0.52 | 2.54 to 4.59 | <0.001 |
| Drive distance between communities in kilometers | −0.0025 | 0.0012 | −0.0047 to −0.0002 | 0.03 |
| | AIC | 215.98 | | |
| | N | 870 | | |
| Post baseline model: After the intervention had taken effect | | | | |
| Intercept | −11.31 | 0.54 | −12.37 to −10.24 | <0.001 |
| Transmission source: control community | 0.90 | 0.51 | −0.11 to 1.90 | 0.08 |
| Transmission type: same community | 2.05 | 0.61 | 0.86 to 3.25 | 0.001 |
| Drive distance between communities in kilometers | −0.0031 | 0.0014 | −0.0059 to −0.0003 | 0.03 |
| | AIC | 137.26 | | |
| | N | 870 | | |

Models were fit to directed opposite-sex HIV-1 transmission pairs identified between ordered pairs of communities in the BCPP trial during the baseline ($n = 51$) and post-baseline ($n = 31$) periods (see methods section Deep-sequence phylogenetics data and Supplementary Note sections S1.1 and S1.2). Post-baseline denotes at least 1 year after baseline household survey activities had concluded in a community (see methods BCPP study description). Coefficients, standard errors and confidence bounds are shown on the linear scale. Two-sided p-values are derived from the Wald Z-statistic ($Z$ = Coefficient/Standard Error). The reference category is transmission between opposite-sex individuals from different communities in the BCPP trial with the source in an intervention community. The intercept describes the risk of transmission (expected probability of viral linkage) in the reference category. Covariates denote the effect on the risk of transmission of (1) a source in a control community, (2) when both the source and recipient reside within the same community and (3) a 1-kilometer increase in the drive distance separating a pair of communities. Risk of same community transmission was approximately 35-fold [13–98] higher at baseline and 8-fold [2–26] higher post-baseline compared to that between individuals residing in different communities (fold-change = exp(coefficient), e.g., 35 = exp (3.56) and 8 = exp (2.05)). Risk of transmission decreased by 27% [95% Confidence Interval (CI): 3–45] per 100 km (27% = 100% × [1 – exp(100 km × −0.0031)]).

could suggest a larger impact of the intervention than originally envisaged[6–8].

We test this hypothesis in one of the four trials, the Botswana Combination Prevention Project. Specifically, we developed a statistical modeling approach that uses directed sexual contacts (transmission pairs) inferred from deep-sequenced HIV virus to estimate the relative extent to which transmissions in trial communities occurred from individuals in the same community; individuals in different communities randomized to the same trial arm; individuals in different communities randomized to the opposite trial arm; and individuals in non-trial communities. In addition, we use our statistical model to estimate the relative benefit of providing the BCPP intervention to all communities nationwide.

## Results

Of the 5114 trial participants who consented to a blood draw for viral genotyping and whose HIV viral whole genomes were successfully deep-sequenced[1,9], 3832 met inclusion criteria for phylogenetic analysis, and from those, we identified 236 clusters of trial participants with genetically similar HIV-1 infections (525/3832 trial participants). Within the 236 genetic similarity clusters we identified 82 directed opposite-sex transmission pairs between ordered pairs of the 30 communities in the BCPP trial (Supplementary Fig. 1), we also identified 71 same-sex pairs between women ($n = 65$) and men ($n = 6$) [9]. Because the transmission of HIV in Botswana and Southern Africa is predominantly through heterosexual contact, and direct transmission is rare between women, we assumed that same-sex pairs represent transmission chains with one or more unsampled intermediates[9–11]. Therefore, we restricted subsequent analyses to the directed opposite-sex transmission pairs. Of the 82 source-recipient pairs, 51 (21 female-to-male, 30 male-to-female) were identified between HIV viral genomes sampled during the baseline period of the trial compared to 31 (16 female-to-male, 15 male-to-female) where the recipient's genome was sampled post-baseline. We defined the post-baseline period as at least 1 year after baseline household survey activities had concluded in a community such that the intervention could have taken effect.

### Relationship between the drive distance separating pairs of communities and the risk of transmission between them
We first demonstrate a relationship between the drive distance separating communities in the BCPP trial and the risk of HIV-1 transmission between them (Table 1 and Fig. 1). We define the risk of transmission as the expected probability of viral-linkage between deep-sequenced HIV viruses of individuals with HIV randomly sampled from their respective communities. Figure 1 and Table 1 show that the risk of transmission decreases as the drive distance separating community pairs increases, specifically by 27% [95% Confidence Interval (CI): 3–45] per 100 km. Note that the decrease in risk of transmission per 100 km is computed from Table 1 as (27% = 100% × [1 – exp(100 km × −0.0031)]). Beyond the effect of distance, the risk of transmission between individuals who reside within the same community was approximately 35-fold [13–98] higher at baseline and 8-fold [2–26] higher post-baseline compared to that between individuals residing in different communities. Note that the fold-change in the risk of transmission is computed from Table 1 as 35 = exp (3.56) at baseline and 8 = exp (2.05) post-baseline.

### Estimating the relative contribution of different sources of infection residing inside versus outside the trial area
Next, using this model to estimate transmissions from communities that were not in the trial, we estimate proportions of transmissions into intervention communities and control communities of the BCPP trial that occurred from individuals in the same community; different communities in the same trial arm; different communities in the opposite trial arm; and non-trial communities (see Supplementary Note sections S1.1 and S1.2). We define non-trial communities as communities outside of the 30 communities that participated in the BCPP trial. We estimated that individuals in non-trial communities accounted for most of the transmissions that occurred to recipients in trial communities, with point estimates ranging from 84 to 92% in intervention communities and 73–92% in control communities (Fig. 2). On average, 90% [95% Confidence Interval (CI): 80–94] of transmissions to recipients in intervention communities and 86% [74–91] of

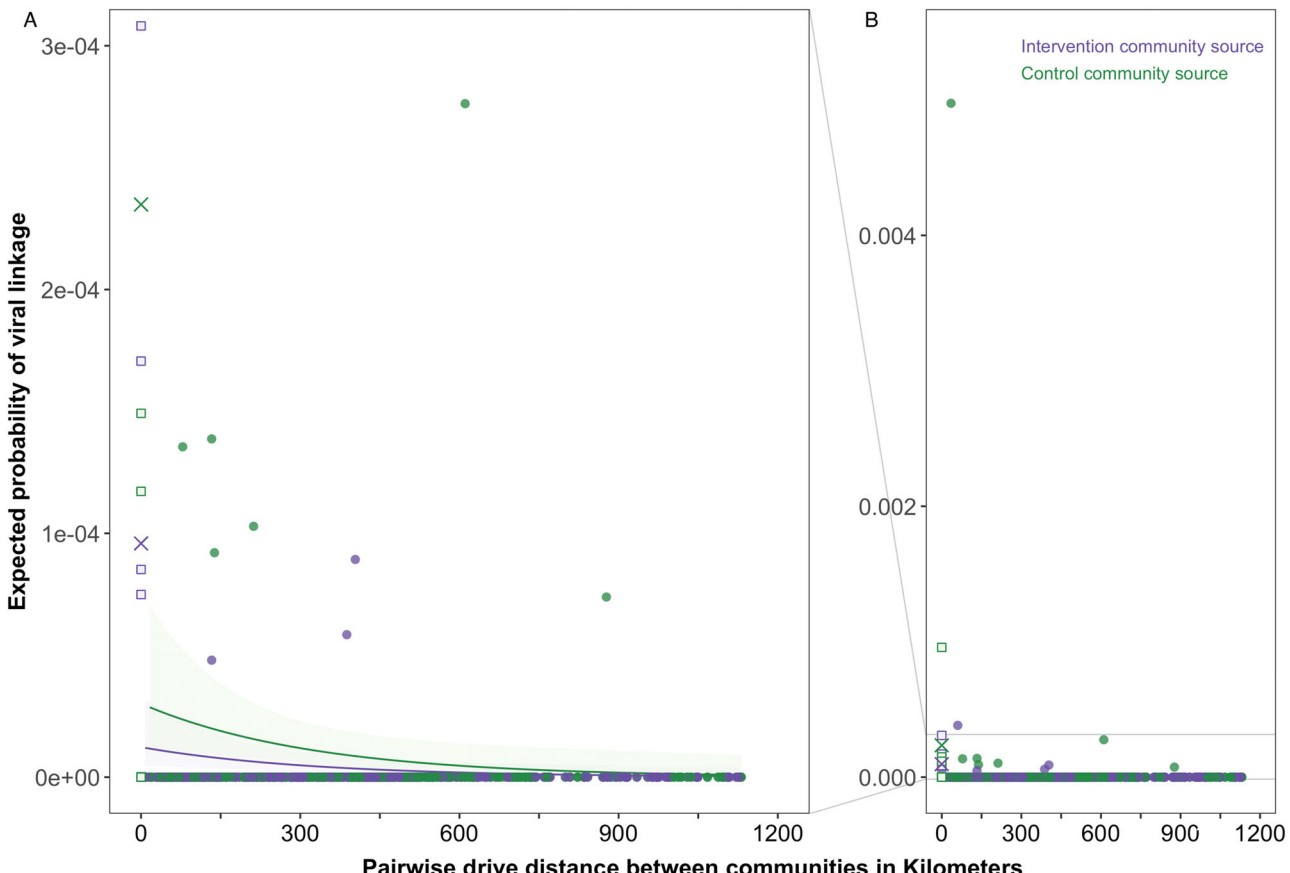

**Fig. 1 | Risk of HIV-1 transmission between communities in the BCPP trial decreases as the drive distance separating them increases.** The plot shows the expected probability of viral linkage, that is, risk of transmission between a pair of individuals randomly sampled from their respective communities in the BCPP trial. The expected probability of viral linkage was predicted with the post-baseline model in Table 1. To improve visibility (**A**) is a zoomed-in plot of the plot in (**B**). Estimates for intervention community sources and for control community sources are shown in purple and green respectively. Solid curves and ribbons (95% confidence interval) show the risk of transmission predicted by the post-baseline model between different communities in the BCPP trial and the associated uncertainty in the estimates. By comparison, solid crosses depict the risk of transmission predicted by the post-baseline model within the same community. Squares and filled circles show the raw data for the 870 ordered community pairs of the 30 BCPP

trial communities (15 intervention, 15 control) that were used to predict the expected probability of viral linkage within the same community (squares) or between different communities (filled circles). Because Digawana intervention community had no participants with successfully sequenced samples during the post-baseline period, community pairs with Digawana as a destination (recipient) community were excluded from the model ($870 = 30 \times 30 - 30$). Among the 870 ordered community pairs, 29 were same community pairs (14 intervention community source, 15 control community source) and 841 were different community pairs (421 intervention community source, 420 control community source). For each of the 870 ordered community pairs, the probability of viral linkage was computed from the raw data as the proportion of directed opposite-sex transmission pairs identified out of the total possible distinct opposite-sex transmission pairs among sampled participants.

transmissions to recipients in control communities were estimated to have sources who lived in non-trial communities (Fig. 3). This finding is consistent with communities in the BCPP trial being densely surrounded by communities outside the trial area and aligns with the fact that the BCPP trial participants represented a relatively small (7.6%) proportion of the national population.

**Proximity to urban centers.** Communities in the BCPP trial are distributed around three major urban areas that each have relatively high numbers of people with HIV; these are Gaborone city in the South-East, Palapye in the Central-East and Francistown city in the North/North-East (Fig. 2 and Supplementary Fig. 2). Figure 2 shows that sexual partners in the same community had a greater impact on transmission in rural communities that are geographically isolated compared to in communities that closely neighbor major urban centers. For example, Gumare intervention community and Shakawe control community in the Northern region of Botswana received an estimated 9% [2–30] and 22% [8–55] of transmissions respectively from individuals in the same community; this estimated percentage was lower for communities on the periphery of densely populated urban areas such as Oodi

intervention community 2% [0.4–7] and Bokaa control community 5% [2–15] in the South-East region. Furthermore, we found that the proportions of transmissions to recipients in intervention communities from individuals in the same trial arm varied across the three major urban areas (Central-East: 6% [2–18], North/North-East: 5% [2–15], South-East: 4% [1–13]) ($\chi^2 = 43, df = 2, P = 4.21E - 10$).

**Impact of communities in the opposite trial arm.** Individuals in control communities contributed a higher proportion of transmissions to intervention communities than the reverse. For example, the proportions of transmissions to recipients in intervention communities from individuals in control communities ranged with point estimates from 4.3 to 5.7%, compared to those to recipients in control communities from individuals in intervention communities that ranged from 1.8 to 2.4% (Fig. 2). On average, 4.8% [4.1–5.0] of transmissions to recipients in intervention communities occurred from individuals in control communities compared to 2.0% [0.6–4.8] of transmissions to recipients in control communities that occurred from individuals in intervention communities, consistent with a benefit of treatment-as-prevention (Fig. 3). Furthermore, Fig. 3 shows that on average 3.2%

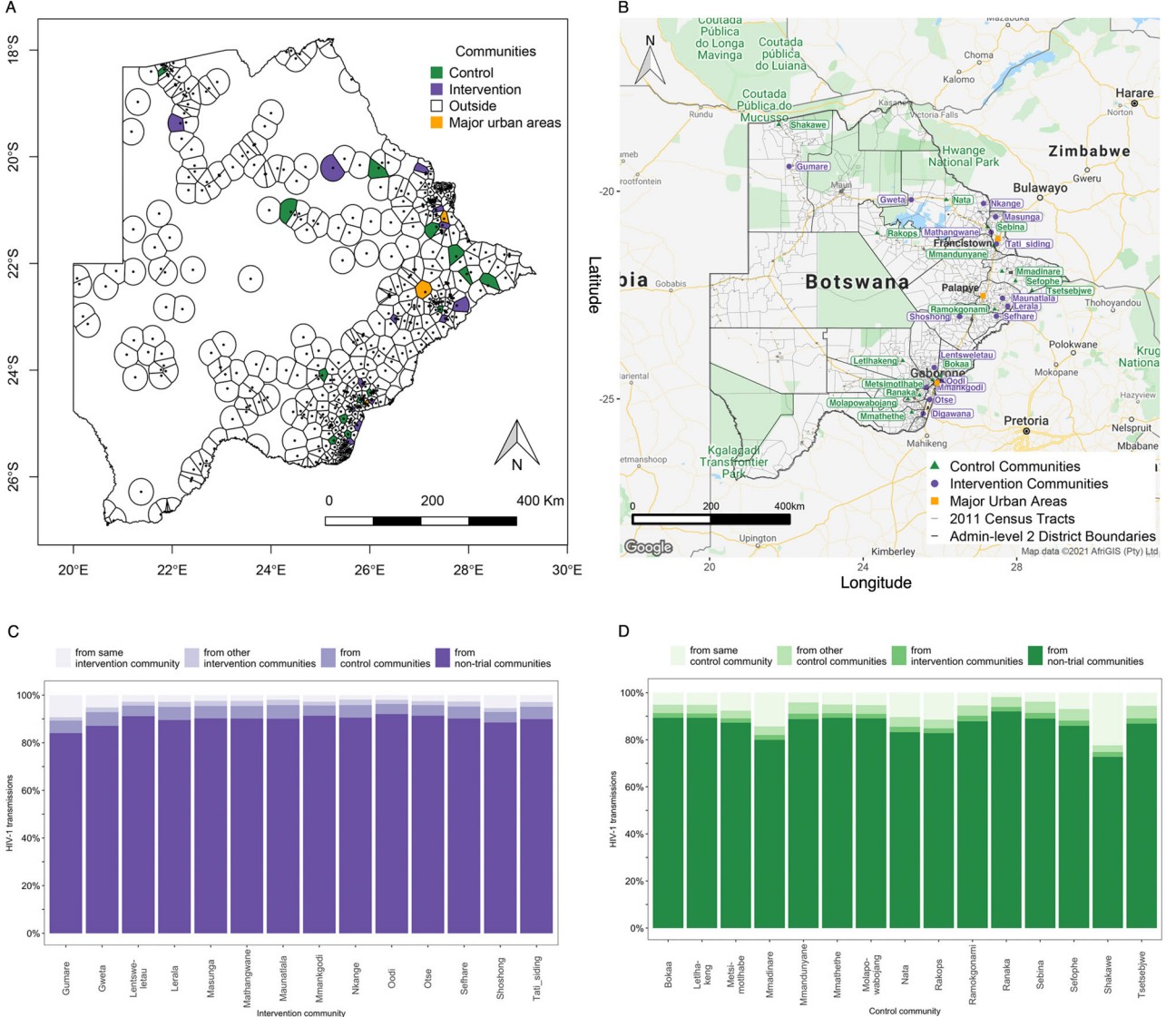

**Fig. 2 | Estimated HIV-1 transmissions into communities in the BCPP trial from different sources of infection. A** A voronoi tesselation map of communities (*n* = 488) in the 2011 Botswana population and housing census showing that communities in the BCPP trial are densely surrounded by communities outside the trial area i.e., non-trial communities. To complement (**A**, **B**) shows the names of the intervention communities and control communities in the BCPP trial within the context of administrative districts and census tracts. **C** shows, in increasing shades of purple, the estimated proportions of HIV-1 transmissions to recipients in intervention communities from individuals in: the same community, other intervention communities, control communities and from non-trial communities. Note that Digawana intervention community is omitted from panel C because there were no successfully sequenced post-baseline samples in the community. **D** shows, in increasing shades of green, the same for recipients in control communities. Most of the transmissions to recipients in BCPP trial communities originated from individuals in non-trial communities.

[0.9–11.5] of transmissions to recipients in intervention communities occurred from individuals in the same community and 1.8% [0.6–4.2] of transmissions occurred from individuals in other intervention communities. In comparison, 7.8% [7.3–24.2] of transmissions to recipients in control communities occurred from individuals in the same community and 3.9% [3.2–4.3] of transmissions occurred from individuals in other control communities.

### Impact of a nationwide intervention

From the post-baseline model in Table 1 we estimate that the relative benefit of a national roll-out of the BCPP intervention would be a 59% [3–87] reduction in transmissions to recipients in trial communities from communities nationwide. For example, we compute the relative benefit of the BCPP intervention from the post-baseline model in Table 1 as (59% = 1 − exp(−0.9)). Then, 95% confidence intervals for the

relative benefit can be computed in two ways. First, we used an empirical bootstrap approach to compute 95% confidence intervals from the 2.5% and 97.5% quantiles of the distribution of 1000 bootstrap samples, 59% [3–87]. Second classical 95% confidence intervals were computed from Table 1 as (lower bound: 1 − exp(−(−0.11)), upper bound: 1 − exp(−(1.90)), 59% [−11 to 85]). Note that the difference in confidence interval estimates provided by the two approaches reflects the sample size of the data, a larger sample size would result in greater agreement between the two approaches. This finding adds evidence that the impact of the BCPP trial intervention could be substantially larger than that observed in the trial if applied nationally.

## Discussion

Global targets set by the Joint United Nations Programme on HIV/AIDS (UNAIDS) to have fewer than 500,000 new infections by the year 2020

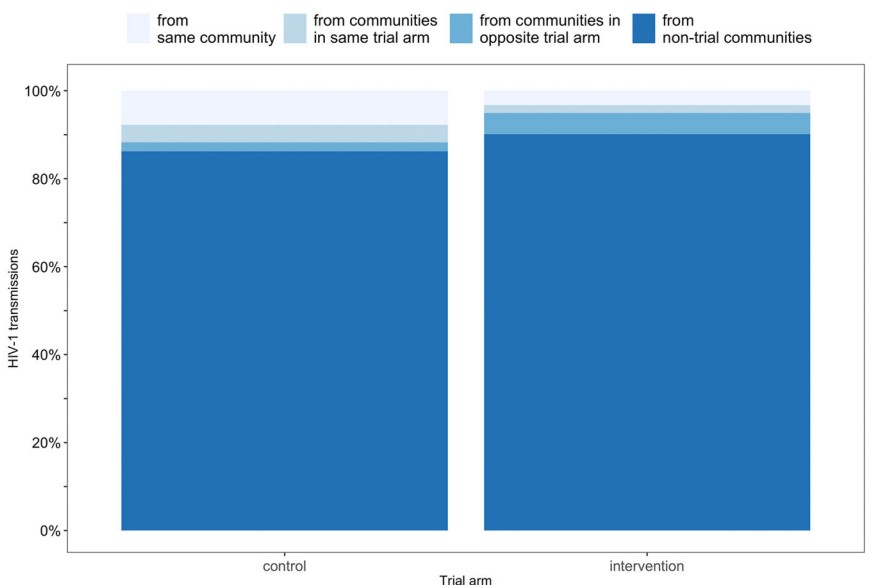

**Fig. 3 | Mean estimates of HIV-1 transmissions that occurred to recipients in intervention communities and control communities in the BCPP trial from different sources of infection.** The barplots show, in increasing shades of blue, the estimated proportions of HIV-1 transmissions to recipients in intervention communities and control communities from individuals in the same community (intervention: 3.2% [95% CI: 0.9–11.5], control: 7.8% [7.3–24.2]), communities in the same trial arm (intervention: 1.8% [0.6–4.2], control: 3.9% [3.2–4.3]), communities in the opposite trial arm (intervention: 4.8% [4.1–5.0], control: 2.0% [0.6–4.8]), and from non-trial communities (intervention: 90.1% [80.1–93.5], control: 86.2% [74.2–91.0]). The mean estimate of the proportion of HIV-1 transmissions to recipients in intervention communities from intervention sources, that is, from individuals in the same intervention community and from individuals in other intervention communities was 5.1% [95% CI: 1.8–15.5].

on a path to reach epidemic control by the year 2030 were missed. Relevant to this, several large community-randomized universal HIV test-and-treat trials that were at the center of HIV prevention efforts in East and Southern Africa showed mixed results[1–5]. To aid interpretation of the complex trial results, inform public health policy decisions about effective HIV prevention strategies, and inform the design of such studies in the future, we developed a statistical modeling approach that uses directed sexual contacts inferred from deep-sequenced HIV to quantify the relative contribution of different sources of infection that might be inside trial communities or outside the trial area. Briefly, to demonstrate the relative extent to which transmissions in intervention communities and control communities of the BCPP trial in Botswana occurred from individuals in the same community; different communities in the same trial arm; different communities in the opposite trial arm; and communities outside the trial area, we first inferred directed opposite-sex transmission events between trial communities using deep-sequence phylogenetics. Then we used the inferred transmission events together with the pairwise drive distances between trial communities and the intervention status of source communities to statistically model the risk of transmission between trial communities. After that we provided pairwise drive distances separating communities that participated in the 2011 Botswana population and housing census and trial communities to the model as input to estimate the risk of transmission (expected probability of viral genetic-linkage had the cases been sequenced) to trial communities from communities nationally. Then, to estimate the number of transmissions into trial communities from all communities nationally we combined estimates of the risk of transmission to recipients in trial communities from communities nationally with population-size estimates from the 2011 Botswana population and housing census[12] and district-level HIV prevalence estimates from the 2013 Botswana AIDS Impact Survey (BAIS 2013)[13] accounting for sex and time period during the trial.

Power analyses and model predictions for the primary endpoints of the BCPP trial in Botswana and the PopART trial in South Africa and Zambia assumed that 20% [95% CI: 15–25] and 5% of sexual partnerships would involve a partner outside one's own community,

respectively[2,8,14]. Strikingly, we found that individuals in non-intervention communities accounted for most of the transmissions that occurred to recipients in intervention communities; with an estimated 90% [80–94] of transmissions attributable to individuals from non-trial communities and 4.8% [4.1–5.0] of transmissions attributable to individuals from control communities. This is underscored by the fact that we identified few of the sources of transmission to individuals in the incidence follow-up cohort from individuals in trial communities. For context, a phylogenetic study that used consensus sequences of the HIV-1 *POL* (polymerase) gene to estimate the relative contribution of local transmission versus external introductions to HIV-1 incidence in the Africa Health Research Institute (AHRI) study population, a rural and peri-urban population located immediately adjacent to the TasP trial study area in KwaZulu-Natal, South Africa, estimated that 35% [20–60] of new infections in the study population were external introductions that occurred from sexual partners outside the study area[5,15]. Most of the external introductions in the AHRI phylogenetics study were estimated to be from sources within the national borders of South Africa with few cross-border external introductions from Botswana, Malawi, Mozambique, Zambia and Zimbabwe (see Fig. 2B in ref. 15). BCPP trial communities closely neighbor three major urban areas in Botswana (Gaborone city in the South-East, Palapye in the Central-East and Francistown city in the North/North-East), and people tend to be fairly mobile in Botswana. By comparison, the AHRI study area is 200 km from Durban, the major urban area in the KwaZulu-Natal province of South Africa. Therefore, it is unsurprising that there would be more external introductions to BCPP trial communities compared to the AHRI study population. In line with the finding from the AHRI phylogenetics study, a clustering analysis conducted by the PANGEA-HIV consortium on HIV-1 viral consensus sequences from the AHRI study population in South Africa, BCPP trial in Botswana, MRC study population in Uganda, PopART study population in Zambia and Rakai study population in Uganda found few clusters including cohorts from different countries. For example, only a single cluster comprising two members was identified between sequence samples from Botswana and Zambia[16]. The limited

cross-border external introductions into Botswana suggest that extending the BCPP trial intervention to all communities nationally to target more sources could effectively reduce the occurrence of new infections.

Accordingly, we estimated that the relative benefit of providing the BCPP intervention to communities nationwide would be a 59% [3–87] reduction in transmissions to recipients in trial communities compared with the 30% reduction that was observed in the BCPP HIV incidence cohort. This was done under an assumption that the intervention effect that was observed when the source individual lived in an intervention community would be similar when extended to a larger geographical area and would reduce the risk of transmission from any individual residing in Botswana, rather than (as in the trial) any two individuals residing in intervention communities. In practice, the intervention effect could vary owing to differences in transmission patterns of different population sub-groups and geographical locations. We note that a larger sample of directed transmission pairs could provide more precise confidence bounds on the estimated relative benefit of the BCPP intervention, however, this was the largest HIV phylogenetics study conducted in Botswana at the time. These findings suggest that substantial reductions in transmission could be achieved if the intervention is applied nationwide and that estimating the relative contribution of various sources of transmission (attributable fraction of cases) could help to guide targeted applications of the intervention where resources are limited. The timing of the implementation of a universal test-and-treat intervention could be crucial. Fast roll-out could limit the spread of infection and shorten the time to reach epidemic control. Furthermore, the estimated reductions in transmissions with a nationwide intervention suggest that the universal HIV test-and-treat intervention could be used as a foundation for incidence reduction upon which other interventions could be layered to close the gap to reach epidemic control.

A key strength of our statistical approach is that we demonstrate how deep-sequence pathogen genomics can be used at scale to assess interventions in cluster-randomized trials of infectious disease prevention. Our analysis is based on the central assumptions that transmission patterns in communities randomized to the control arm of the trial are representative of those found in non-trial communities, and that, the population-size and HIV prevalence of communities are known; and that the HIV prevalence in administrative districts is representative of that in communities (see Supplementary Note section S1.1 Population-based molecular source attribution model). There are some limitations to our analysis: First, our statistical approach is informed by pairwise drive distances separating pairs of communities and could be improved with mobile phone data to gain insight on daily commutes and seasonal migration for work (for example: farming and mining) and holidays. Second, HIV viral sequences of cases were collected only in trial communities. However, Fig. 1 and Table 1 show a relationship between the drive distance separating pairs of communities in the BCPP trial and the estimated risk of transmission between them. This relationship allows us to use the drive distances separating trial communities and non-trial communities to estimate the expected probability of viral linkage to source cases in non-trial communities had cases in non-trial communities also been sequenced. Third, our model assumes that the intervention only prevents transmissions from intervention sources, that is, people with HIV in intervention communities. In practice, the intervention averts also some portion of transmissions to recipients in intervention communities through voluntary safe male circumcision[11]. Accounting for the impact of male circumcision could result in even more averted transmissions. In addition to this mechanistic point, we also note that both the RCT estimate and the estimate in the present analysis contain significant uncertainty, with considerable overlap in the confidence intervals of the maximum preventable infections 14% [9–26] and the estimate in the RCT of the proportion prevented 30% [10–54]. Note that the

estimate of preventable infections is computed from the estimate that 86% [74–91] of transmissions in control communities occurred from non-trial communities as 14% = 100% − 86%. We used the product of district-level HIV prevalence estimates from the 2013 Botswana AIDS Impact Survey (BAIS 2013)[13] and community-level population sizes from the 2011 census, at the time the most recent estimates, to scale the potential contribution of each location to transmission, representing a simplifying assumption that the relative size of the transmitting population (sex-specific) in each location during the trial period was equal to the relative size of the estimated number of people with HIV (sex-specific) from the product of these two numbers[12]. Our model assumes a similar risk of transmission from male and female transmitters. In practice, there is likely to be heterogeneity in the risk of transmission by sex and age of the transmitter. To broaden insights, our statistical modeling approach could be applied to estimate the relative contribution of various sources of infection by age and sex in the other community-randomized universal HIV test-and-treat trials that have assembled deep-sequence genomic data, for example the PopART trial in South Africa and Zambia. To augment sample size and account for the impact of same-sex transmission in Botswana, a future study could include directed same-sex transmission pairs in the model of the risk of transmission between communities. Out of the 153 highly supported probable transmission pairs identified in BCPP, there were 82 directed opposite-sex pairs and 71 same-sex pairs (65 female-female, 6 male-male)[9]. We restricted our analysis to opposite-sex pairs because of insufficient information to identify the number and sex of unsampled intermediates in same-sex pairs at the time of analysis and reserve the inclusion of same-sex pairs for future study[9,17]. Our model could be improved further by accounting for other HIV prevention interventions such as male circumcision, condoms and pre-exposure prophylaxis. That said our parsimonious model provides helpful insight into the impact of the BCPP intervention in reducing transmissions. We cannot rule out possible bias in the use of proviral DNA for the inference task of who infected whom, however, the successful use of proviral DNA in viral evolution studies suggests such bias could be limited. Future studies could systematically compare proviral DNA and viral RNA for the inference task of who infected whom.

Our findings have implications for public health policy and for the design of effective HIV prevention strategies. By deconstructing the relative contribution of different sources of infection in intervention communities versus control communities this work aids interpretation of the complex universal HIV test-and-treat trials in which the intervention is administered on one group (people with HIV) and the outcome (reduction in number of new cases) is measured on another group (people in the same community without HIV). For example, our findings elucidate the potential impact of a nationwide intervention and provide insight on the extent to which the BCPP intervention was diluted by spillover infections from control communities and from communities outside the trial area. Furthermore, our findings inform on-going public health policy discussions on whether the HIV testing component in national HIV prevention programs should be centered on facility-based testing at clinics and index-based testing of family and sexual contacts of people with HIV or anchored on intensive universal household-based HIV testing as was done in the trials. For example, this study shows how the combination of universal household-based HIV testing and routine HIV testing in health facilities - as was done in the combination prevention intervention in the BCPP trial - allows us to infer transmission patterns within and between communities to guide HIV prevention strategies. Also, Wirth et al. show that new diagnosis of people with HIV were higher with household-based testing compared to venue-based testing at mobile clinics. As highlighted in the BCPP study description in the methods, the BCPP intervention was deployed through a highly decentralized system of health care facilities that are overseen by District Health Management Teams (DHMTs) of health care workers in the Ministry of Health, therefore, a nationwide roll-out

of the BCPP intervention could be administered in the same way[11,18,19]. An initial nationwide intervention to maximize capture of sources of transmission could be done once followed by regular targeted interventions informed by genomic surveillance.

In sum, this work shows that individuals residing in communities outside the BCPP trial area accounted for most of the transmissions to recipients in intervention communities, limiting the impact of the BCPP trial intervention. Furthermore, substantial gains in reducing transmission could be made with a nationwide application of the intervention. With the introduction of interventions at the community-level (universal test-and-treat) and individual-level (pre-exposure prophylaxis and self-testing) our analysis suggests that genomic surveillance could provide a crucial platform to assess interventions allowing us to track how pathogens spread and evolve overtime in response to different interventions. For example, samples collected for routine viral load testing could also be sequenced to track the directional spread of infection within and between communities and between age-sex population sub-groups. This could help to identify population sub-groups and communities in which HIV prevention interventions need to be further strengthened to reduce transmission. Pairing genomic information with information from studies that explicitly quantify the impact of social behavioral change on interventions could aid the interpretation of HIV prevention studies and evidence-based policy design. Based on our findings, we recommend that studies of infectious disease prevention consider the impact of sources of transmission beyond the reach of the intervention when evaluating interventions to inform public health programs.

## Methods

### BCPP study description and data
**BCPP study description.** The Botswana Combination Prevention Project (BCPP, also known as the Ya Tsie trial) was a pair-matched community-randomized trial to evaluate the effect of universal HIV testing and treatment on HIV incidence reduction. The trial was conducted from 2013–2018 in 30 rural and peri-urban communities distributed across Botswana[1,20]. The BCPP intervention was administered through an existing system of highly decentralized health care facilities and personnel overseen by DHMT's of health care workers in the Ministry of Health. Trial participants were adults aged 16–64 years and the average population size eligible to participate in each trial community was 3820 people. Communities were matched into 15 pairs based on three criteria: geographical proximity to major urban areas (Gaborone city, Palapye and Francistown city), population size and age structure, and access to health services; then within each pair, communities were randomized into the intervention and control arms of the trial. The 15 intervention communities in the trial received expanded access to universal HIV testing (with attempt to test all willing adult residents who did not have documented positive HIV status), strengthened linkage-to-care for early treatment, and expanded treatment availability. After a period of community sensitization through door-to-door canvassing, community leadership engagement and public loudspeaker announcements, mobile and home-based HIV testing campaigns were conducted within each intervention community over approximately two consecutive months[11]. Routine testing in intervention community health facilities was reinforced to diagnose all people with HIV and avail them early treatment. An additional effort was made to offer HIV testing to men and youth where they work and socialize, for example: at bars and community football (soccer) matches. To strengthen linkage-to-care, people with HIV who were not on treatment were assisted to schedule an appointment at a local clinic, provided text alerts prior to the appointment and followed-up to reschedule in the case of a missed appointment. Access to services for safe male circumcision and prevention of mother-to-child transmission was also expanded in intervention communities. By comparison, control communities received the standard-of-care, which before 2016

meant that people with HIV qualified to start antiretroviral treatment when their CD4 cell count was below 350 cells per microliter. Beginning June 2016, the national HIV treatment policy was changed to universal treatment meaning that immediate antiretroviral treatment was now available in both arms of the BCPP trial. The first-line regimen, provided by the Government of Botswana to all trial communities, also changed from efavirenz (EFV)-tenofovir disoproxil fumerate (TDF)-emtricitabine (FTC) to dolutegravir (DTG)-TDF-FTC[1]. To evaluate HIV incidence reduction, an HIV incidence follow-up cohort was established through a baseline household survey of a random sample of 20% of households in each trial community. Annual household surveys with retesting for HIV (in persons who were HIV-negative at the last survey) were then conducted in the same 20% household sample in all 30 communities during the trial. The BCPP trial comprised 7.6% (175,664) of the national population and showed a 30% reduction in the occurrence of new HIV infections in intervention communities compared to control communities over an average of 29 months[1]. In addition, the BCPP trial conducted an end-of-study survey of 100% of households in 3 intervention communities and 3 control communities to assess progress on the 90-90-90 UNAIDS targets. Trial participants with HIV were invited to provide a sample for viral phylogenetic analysis. This included all people with HIV from (1) the baseline household survey, (2) annual household surveys, (3) end-of-study survey, as well as (4) all people with HIV (but not yet on ART) who were referred for treatment during community-wide testing and counseling campaigns, (5) all people with HIV that later presented at health care facilities in intervention communities and (6) all people with HIV who were already receiving HIV care at health facilities in intervention communities.

**Deep-sequence phylogenetics data.** Near full-length genome sequences were obtained using predominantly proviral DNA (as the majority of study participants were virally suppressed on ART) or RNA. The HIV-1 viral consensus whole genomes of individuals that met minimum criteria for inclusion in phylogenetic analyses were ones that had fewer than 30% of bases missing beyond the first 1000 nucleotides ($\geq 6{,}300$ nucleotides available) (see criteria for inclusion in phylogenetic analyses in ref. [9]). To efficiently use computational resources, viral consensus whole genomes were used to identify groups (or clusters) of trial participants with genetically similar HIV-1 infections as a filtering step to exclude distantly related sequences from deep-sequence phylogenetic analysis[9]. We defined clusters of genetically similar HIV-1 infections as groups of two or more trial participants whose viral whole-genome consensus sequences were separated by a genetic distance less than 4.5% nucleotide substitutions per site. This empirical threshold of 4.5% substitutions per site was motivated by the distribution of genetic distances separating subtype C HIV-1 viral whole-genome consensus sequences from epidemiologically linked couples in the HIV Prevention Trials Network (HPTN) 052 study[21,22] (see Consensus sequence phylogenetics to identify clusters of participants with genetically similar HIV-1 infections in ref. [9]). A detailed description of the deep-sequence phylogenetic analysis is published in ref. [9]. Briefly, we performed parsimony-based ancestral host-state reconstruction with the phyloscanner software[23,24] to identify pairs of trial participants with genetically similar HIV-1 infections and the probable direction of transmission between them. Based on empirical data we defined transmission pairs with strong support for phylogenetic linkage and direction of transmission, that is highly supported pairs, based on a phylogenetic linkage and direction of transmission score threshold of 57% (see section Identifying probable source-recipient pairs with strong phylogenetic evidence for linkage and direction of transmission in ref. [9]). We identified both highly supported same-sex and opposite-sex (female-to-male or male-to-female) transmission pairs, however, we restricted our analyses to opposite-sex pairs because we could not reliably distinguish between direct transmission in same-sex pairs, and same-sex members of transmission chains with

unsampled intermediates[9]. For brevity, we refer to the directed opposite-sex transmission pairs as source-recipient pairs. For each identified transmission pair there was accompanying metadata on the names of the communities in which the source and recipient partners reside and the randomization-condition to which their communities were assigned.

**Pairwise driving distance data.** Pairwise drive distances between ordered pairs of 488 communities in the 2011 Botswana population and housing census were successfully sourced from the google distance matrix application programming interface (API) with the mapsapi package v0.5.0 in R v4.1.1[25]. Note that the 2011 census was the most recent census at the time of the trial. The 488 census communities included all 30 communities in the BCPP trial. Therefore, of the possible 488 × 488 ordered community pairs between the 488 census communities we sourced 488 × 30 ordered community pairs that had any of the 488 census communities as a source (origin) community and any of the 30 trial communities as a recipient (destination) community.

**Population-size and HIV prevalence estimates.** Population-size estimates of 488 communities in the 2011 Botswana population and housing census were sourced from ref. 12, and district-level HIV prevalence estimates were obtained from the 2013 Botswana AIDS Impact Survey (BAIS 2013)[13]. Note that BAIS 2013 provided the most recent sub-national estimates for HIV prevalence at the time of the trial.

### Estimating transmissions to recipients in BCPP trial communities

To estimate transmissions that occurred to recipients in trial communities from different sources of infection nationally, we first used viral sequence samples from the 30 BCPP trial communities and deep-sequence phylogenetics to infer probable directed (opposite-sex) transmission events between the trial communities. Then we used the inferred transmission events to statistically model the risk of transmission between trial communities in a negative-binomial regression framework. In the model the inferred transmission events served as the response variable and the following variables were predictors: (1) the pairwise drive distance separating the source and recipient communities, (2) whether the source community was randomized to receive the intervention, and (3) whether the source community and the recipient community were the same (within-community transmission) or different (between-community transmission). After that we used the pairwise drive distances between the 488 communities in the 2011 Botswana population and housing census and the 30 trial communities as input to the model of the risk of transmission between trial communities to predict the risk of transmission (expected probability of viral genetic-linkage had the cases been sequenced) to trial communities from communities nationally. Finally, to estimate the number of transmissions into trial communities from all communities nationally, estimates of the risk of transmission to recipients in trial communities from communities nationally were combined with population-size estimates from the 2011 Botswana population and housing census[12] and district-level HIV prevalence estimates from the 2013 Botswana AIDS Impact Survey (BAIS 2013)[13] accounting for sex and time period during the trial. The supplementary note provides a detailed account of the statistical approach used to estimate the relative contribution of different sources of infection in: (1) the same community, (2) different communities in the same trial arm, (3) different communities in the opposite trial arm and (4) in communities outside the trial area.

### Ethics statement

The BCPP study was approved by the Botswana Health Research and Development Committee and the institutional review board of the Centers for Disease Control and Prevention; and was monitored by a data and safety monitoring board and Westat. Written informed consent for enrollment in the study and viral HIV genotyping was obtained from all participants.

### Reporting summary

Further information on research design is available in the Nature Portfolio Reporting Summary linked to this article.

## Data availability

All relevant data are within the paper, figures and tables. Epidemiological data. Population-size estimates were sourced from ref. 12 and district-level HIV prevalence estimates were sourced from the 2013 Botswana AIDS Impact Survey (BAIS 2013)[13]. Additionally, both can be accessed at: https://github.com/magosil86/spillover-infections. Driving distance data. Pairwise drive distances were sourced from the google distance matrix application programming interface (API) with the mapsapi package v0.5.0 in R v4.1.1[25] and can be accessed at: https://github.com/magosil86/spillover-infections. Sequence data. HIV-1 viral whole genome consensus sequences are accessible as a Dryad dataset (https://doi.org/10.5061/dryad.0zpc86706). HIV-1 reads are available on reasonable request through a concept sheet proposal to the PANGEA consortium. Contact details are provided on the consortium website (https://www.pangea-hiv.org). Counts of transmission pairs inferred from sequence data were sourced from ref. 9 and can be accessed at: https://github.com/magosil86/spillover-infections.

## Code availability

A code repository has been made available at the following URL: https://github.com/magosil86/spillover-infections [26]. The code is also available via Zenodo at: https://doi.org/10.5281/zenodo.17641168.

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

## Acknowledgements

We are grateful to participants and collaborators from the Botswana Combination Prevention Project for their support during this work. We also thank the following colleagues for their helpful suggestions: Stephanie Marie Davis, Carol A. Ciesielski, Anindya De and Stacie Greby. This study was supported by the National Institute of General Medical Sciences (U54GM088558) M.L; the Fogarty International Center (FIC) of the U.S. National Institutes of Health (D43 TW009610) L.E.M; and the President's Emergency Plan for AIDS Relief through the Centers for Disease Control and Prevention (CDC) (Cooperative agreements U01 GH000447 and U2G GH001911) L.E.M, J.M, P.B, R.L, M.P, S.L, M.E, NIH K24 AI131928 S.L as well as the Morris-Singer Fund, the VK Fund for the Harvard Center for Communicable Disease Dynamics and the Bill and Melinda Gates Foundation. The findings and conclusions in this report are those of the author(s) and do not necessarily represent the official position of the funding agencies.

## Author contributions

L.E.M., V. D., M.L., S.L., M.E. conceived the study; M.E., S.L., J.M., M.L., V.N., M.P., P.B., J.M., K.E.H., T.G., S.M., R.L., R.G. facilitated data acquisition; L.E.M. performed statistical and computational analyses; M.L., V.D., C.F., E.T. evaluated statistical and computational analyses; S.L., M.P. administered the project; M.L., V.D., S.L., M.E. supervised the project; L.E.M., M.L., V.D. wrote the original draft of the manuscript; all authors reviewed and approved draft and final versions of the manuscript.

## Competing interests

Consulting fees. M.L.: Merck, University of Virginia Miller Center, Janssen. Unpaid consulting. M.L.: Janssen, Pfizer, Astra Zeneca; Payment or honoraria for lectures, presentations, speaker's bureaus, manuscript writing or educational events. M.L.: Sanofi Pasteur, Bristol Myers Squibb; Leadership or fiduciary role in other board, society, committee or advocacy group, paid or unpaid. S.L.: Member, Finance Board; member, Board of Directors; both for the Botswana Harvard AIDS Institute Partnership (unpaid). The remaining authors declare no competing interests.

## Additional information

[1]Center for Communicable Disease Dynamics, Department of Epidemiology, Harvard T.H. Chan School of Public Health, Harvard University, Boston, MA, USA. [2]Wellcome Sanger Institute, Cambridge, UK. [3]Department of Statistics, The Wharton School, University of Pennsylvania, Philadelphia, PA, USA. [4]Harvard T.H. Chan School of Public Health AIDS Initiative, Department of Immunology and Infectious Disease, Harvard T.H. Chan School of Public Health, Harvard University, Boston, MA, USA. [5]Botswana Harvard AIDS Institute Partnership, Gaborone, Botswana. [6]Division of Global HIV/AIDS and TB, Centers for Disease Control and Prevention, Atlanta, GA, USA. [7]Ministry of Health, Republic of Botswana, Gaborone, Botswana. [8]Oxford Big Data Institute, Li Ka Shing Center for Health Information and Discovery, Nuffield Department of Medicine, Old Road Campus, University of Oxford, Oxford, UK. [9]Department of Biostatistics, Harvard T.H. Chan School of Public Health, Harvard University, Boston, MA, USA. [10]Division of Infectious Diseases and Global Public Health, University of California San Diego, La Jolla, CA, USA. [11]Division of Infectious Diseases, Brigham and Women's Hospital, Boston, MA, USA. [12]Division of Biostatistics, Herbert Wertheim School of Public Health and Human Longevity Science, University of California San Diego, La Jolla, CA, USA. [13]Present address: Department of Medicine (Infectious Diseases) and Department of Biology, Stanford University, Stanford, CA, USA. [14]These authors jointly supervised this work: Shahin Lockman, Max Essex, Victor De Gruttola, Marc Lipsitch. ✉e-mail: lmagosi@hsph.harvard.edu; lipsitch@stanford.edu

