## [Transparent Peer Review file · Nature Communications]

Unpacking sources of transmission in HIV prevention trials with deep-sequence pathogen data

Corresponding Author: Dr Lerato Magosi

Version 0:

Reviewer comments:

Reviewer #1

(Remarks to the Author)

The authors present an interesting and worthwhile study of the potential drivers for the varied success of test-and-treat HIV trials, using a new statistical approach integrating HIV sequence and drive distance data with traditional epidemiological information.

Applying this approach to the Botswana Combination Prevention Project (BCPP), the authors make the noteworthy finding that an estimated 90% [CI 81-93%] of new HIV infections that occurred in communities receiving combination prevention including test-and-treat originated from communities outside the trial area. This has far reaching implications for the design and interpretation of related trials as well as public health policy more broadly.

I have comments regarding the extent to which available data support the above estimate, as well as minor comments that relate to the data set and interpretation of findings, which I hope once addressed will improve what is a creative and compelling manuscript.

Major comments

The BCPP reported a more than 30% reduction in new HIV infections in the intervention group relative to the standard-care group (Makhema et al. 2019, DOI: 10.1056/NEJMoa1812281); how can this reduction be explained alongside the author's estimate here that 85.6% [CI 73.6 - 90.5%] of transmissions in the standard-care group arose from non-trial communities?

There would, assuming the above, be a subset of new HIV infections (14.4% [CI 9.5 - 26.4%]) for which a local intervention could theoretically have an impact. If the intervention was completely successful, the subset could reduce completely to give an overall reduction in new HIV infections of at most 14.4%; this does not reach the more than 30% reduction observed (Makhema et al. 2019) and makes the above estimate of 85.6% seem untenable.

Secondly, the counterfactual modelling estimate that a nationwide application of the intervention could reduce transmission by 59% [CI 3-87%] has considerably wider confidence intervals than one could reasonably expect; as an example, given that combination prevention including test-and-treat has been shown as able to reduce transmission by more than 30% for an intervention group, as described (Makhema et al. 2019), there should be a strong prior that the intervention would have the same or greater impact when implemented nationally ($\geq 30\%$), due to transmission from neighbouring communities being reduced relative to localised interventions.

Can the authors explain how a nationwide application of the same intervention at the time of the BCPP could reasonably reduce HIV transmission by as little as 3%, as reflected in the above confidence intervals?

Further, if there are 'limited cross border introductions into Botswana', could the authors comment on why a nationwide application of combination prevention including test-and-treat could not reduce new HIV transmissions to a greater extent than estimated?

Minor comments

An introductory sentence detailing the available treatments and the standard of care during the study period would be beneficial, particularly as readers return to this manuscript in years to come.

The rates of directed transmission appear lower than one could expect; the authors report that 82 directed opposite-sex transmission pairs were identified, representing 2.1% (82/3832) of included sequences. Could the authors comment on this, potentially referencing previous work where relevant parameters have been identified or approaches calibrated?

A reader may assume that same-sex transmission is negligible in Botswana on reading the manuscript; transmission pairs are only classified as heterosexual, with directionality as male-female or female-male, which contributes to the source attribution model and many findings from the manuscript. I do not believe that this was intended by the authors, although I believe that careful wording is required both in the manuscript methods and discussion to not reinforce cultural norms that may stigmatise and erase same-sex relationships in country; the authors could reference their own work in this area (Magosi et al. 2022), including the challenges of appropriately describing the inferred transmission between women in southern Africa, as well as other relevant works (Matlapeng et al. 2023, PMID: 37830348) to address this.

Confidence intervals could be a worthwhile addition to Figure 4, given that the nationwide intervention estimates have a wide confidence overall.

There may need to be a purple confidence interval for the curve in Figure 1A.

The authors could comment on the practicalities of implementing a nationwide introduction, as well as the estimated benefits if achieved; are there limitations on staffing or resources that make localised introductions for feasible, and what would key measures of success be for the intervention given co-occurring changes to the standard care.

The finding that partners in the same community contributed more to transmission in rural communities that are geographically isolated (Lines 240-242) seems guaranteed by the model, given the use of pairwise travel distance data. Is there a hypothetical scenario where this would not be the case?

Dr George Taiaroa

(Remarks on code availability)

The code seems readily available, although I will not have time to replicate the analyses here.

Reviewer #2

(Remarks to the Author)

This study provides a unique approach to evaluate the effect of population-level HIV prevention interventions on HIV transmission and a modelling approach to estimate potential benefits of intervention scale-up to a national level. The approach combines phylogenetic inference and statistical modeling. The findings have substantial public health importance and highlight the advantage of broader nation-wide interventions over localized ones. While the paper is very compelling, a few important limitations need to be addressed before publication.

Major comments:

1. Given that the number of non-trial communities is much higher than trial communities (458 vs 30), it is expected by chance that those communities will contribute the majority of infections coming from the outside. I suggest permutation analysis with random assignment of "trial" status to communities, to allow generation of baseline expectation of contributions from non-trial communities.
2. The modelling study assumes that all transmissions happen within the country and disregard foreign importations, which might have limited the estimated effect of a potential nationwide intervention.
3. Why were only opposite-sex couples considered? Even in the case of underreporting of MSM contacts or drug use contacts, were same-sex pairs not discovered through the phylogenetic explorations? Would be great account for behavioral risk factors in the model.
4. I don't think that "transmitter's" gender was taken into account, which assumes equal probability of infection for both and might not be a fair assumption. Considering both biological and epidemiological factors that might make infections from one gender to the other more likely might be important for population-level transmission inference.
5. How do the authors account for the fact that proviral DNA may contain archived, non-infectious virus that does not reflect actively transmitted virus? Could using proviral DNA have introduced bias in estimated transmission links in the analysis?

Minor comments:

1. "Source of infection" when referring to individuals or populations can be problematic; suggest using non-stigmatizing language as "infections attributed to" or "possible lineage exporting location".
2. Some of the results are overstated, as the observed reduction is within the CI of the modelled scenario (line 44-45)
3. Was there a cutoff to determine "genetically similar viruses?" (line 138)

(Remarks on code availability)

Reviewer #3

(Remarks to the Author)

(Remarks on code availability)

Reviewer #4

(Remarks to the Author)

Unpacking sources of transmission in HIV prevention trials with deep-sequence pathogen data BCPP/Ya Tsie study NCOMMS-24-55987-T

Overall:

This study estimated the relative contribution of different sources of HIV infection in a community randomized trial of HIV prevention, conducted from 2013 to 2018, and concluded that 90% of new infections that occurred in individuals residing in intervention communities likely originated from individuals residing in communities outside of the trial area. As the authors note, the findings “provide insight on the extent to which the BCPP intervention was diluted by spillover infections from control communities and from communities outside the trial area.” This dilution effect adds to the challenge of contamination that occurs in implementation science studies when the control communities are also exposed to the intervention. The related concepts of dilution and contamination in implementation science studies are well-known factors that have the potential to undermine the accurate assessment of an intervention’s impact [Boily MC et al., 2012]. In this analysis, however, the authors have taken these concepts one step further by using deep sequencing data and statistical analyses (especially of geographic distances) to quantify the role of dilution by spillover in the Ya Tsie study. Since the Discussion section notes spillovers of far less than 90% in other studies, the paper implicitly acknowledges that the high level of spillover in Ya Tsie does not necessarily generalize to other settings at other points in time. The authors instead suggest that other prevention implementation science studies should also use similar deep sequencing and statistical methods to estimate the ‘true’ impact of a community intervention. If these methods require substantial financial resources, however, the concluding point may seldom be feasible.

Comments/Questions:

1. Matches in sequencing data: It’s clear these were not perfect matches but how often did recipient sequences ‘match’ with only one source sequence? How often were there zero ‘matches’, or more than one ‘match’?
2. Selection bias:
 - a. Lines 124-130 describe which people living with HIV from the trial were invited to join the study by providing a sample for viral phylogenetic analysis but were any excluded? And of those invited, what percentage agreed?
 - b. The sample size in the end was large at 3,832 and all were from trial participants. About 25% of consenting participants had samples that were not successfully sequenced (3832/5114). Did the authors look at the demographic characteristics of those whose samples were sequenced versus (i) those who did not consent) and (ii) had consented but had samples that could not be sequenced? A comparison of the characteristics and how similar or different they are would help alleviate concern for selection bias.
3. Pre-/post-2016: It will strengthen the paper if the authors were stratified analyses by pre-/post- 2016, when test-and-treat became a national strategy. Did the contribution of transmission from non-trial sites become less pronounced post 2016? One would surmise that would be the case after widespread implementation of test-and-treat. If the authors can demonstrate this, their causal argument would be strengthened. There may be sample size issues for these stratified analyses but a sensitivity analysis on this would be helpful.
4. Urban/rural analysis: The authors indicated that in two rural communities, transmission within the same community vs. transmission across different communities was more common because they were isolated and far away from the three urban centers. Did this phenomenon occur in other communities? The authors did not present a formal urban/rural stratified analysis which could strengthen the causal claim.
5. Key populations: There was no discussion on key populations, particularly men-to-men transmission. A mention of why this was not included would be helpful, given the outsized role key populations play in ongoing transmission.

Minor comments:

1. “Pairwise drive distance”: it would be more intuitive to call this “Pairwise driving distance” and then explicitly define it in the Methods section (e.g., at approx. Line 148) as the estimated driving distance between each of 488 census communities and the 30 trial communities.
2. One wonders if, instead of 2011 census data and 2013 AIDS indicator survey data, it would have made a difference in the results if the authors had been able to use data from time periods closer to the end date (2018) of the Ya-Tsie study.
3. Table 1: Rather than showing how coefficients are converted to % on page 12 in the text itself, the authors might consider including these calculations directly in Table 1 to help readers with the flow of the text on page 12 and with the interpretation of Table 1.

Reference:

Boily MC, Mâsse B, Alsallaq R, Padian NS, Eaton JW, Vesga JF, Hallett TB. HIV treatment as prevention: considerations in the design, conduct, and analysis of cluster randomized controlled trials of combination HIV prevention. PLoS medicine. 2012 Jul 10;9(7):e1001250.

(Remarks on code availability)

N/A

Reviewer #5

(Remarks to the Author)

(Remarks on code availability)

N/A

Version 1:

Reviewer comments:

Reviewer #1

(Remarks to the Author)

The authors have comprehensively and satisfactorily addressed my previous comments. I have no additional remarks, and commend them on the work.

(Remarks on code availability)

The code appears readily accessible although I do not have scope to recreate the analyses here.

Reviewer #2

(Remarks to the Author)

We appreciate the clarifications and additional explanations provided. The manuscript was improved following the recommendations from all reviewers, but we feel a few important points still need more attention. In particular, assumptions about how "transmitter's" gender was handled in the current model, and the potential bias from using proviral DNA should be stated more openly as limitations so readers can better interpret the results. We outline these points below:

14. I don't think that "transmitter's" gender was taken into account, which assumes equal probability of infection for both and might not be a fair assumption. Considering both biological and epidemiological factors that might make infections from one gender to the other more likely might be important for population-level transmission inference.

Author's response:

The reviewer raises a good point. In a previous study we estimated the risk of transmission between trial communities by age and sex accounting for sampling variation (Please see figure 12 in [10]).

Reviewer's Comment:

Thank you for referring to your earlier study. However, the concern relates specifically to the current model that includes both trial and non-trial communities. It is still unclear if "transmitter's" gender was incorporated into this model, or if male-female and female-male transmissions were modeled as equally likely.

15. How do the authors account for the fact that proviral DNA may contain archived, noninfectious virus that does not reflect actively transmitted virus? Could using proviral DNA have introduced bias in estimated transmission links in the analysis?

Author's response:

We thank the reviewer for their comment and would like the reviewer to kindly note that the utility of proviral DNA for the analysis of HIV transmission dynamics and for the surveillance of transmitted drug resistance has been successfully demonstrated in multiple studies (PMID: 26041893).

Reviewer's Comment:

Thank you for the provided reference. However, that study used proviral DNA sequences to determine phylogenetic clustering rather than directionality of infection, so it does not address the concern here. Would the timing of viral suppression bias the analysis of deep-sequencing data in phyloscanner? Could you cite any validation of proviral DNA vs. RNA (active) virus for the specific inference task (who infected whom)? Or discuss the potential bias from archived proviral DNA and acknowledge as a potential limitation in the manuscript.

We also believe that the response to Reviewer 4's comment number 20b is not satisfactory, as the same analysis

(comparing demographic characteristics of those whose samples were included in the analysis vs those who were not) could be helpful to demonstrate lack of bias in sample selection. This does not suggest “including sequences with greater missingness” as authors interpreted, but would allow better context of how included and excluded sequences differ.

(Remarks on code availability)

Reviewer #3

(Remarks to the Author)

(Remarks on code availability)

Reviewer #4

(Remarks to the Author)

The authors have responded to all of the reviewers' comments. I have no further comments for the authors.

(Remarks on code availability)

Reviewer #5

(Remarks to the Author)

(Remarks on code availability)

Author responses for Nature Communications: NCOMMS-24-55987-T

Note: Reviewer comments are numbered from 1 through 24 and italicized for ease of reference.

REVIEWER COMMENTS

Reviewer #1 (Remarks to the Author):

The authors present an interesting and worthwhile study of the potential drivers for the varied success of test-and-treat HIV trials, using a new statistical approach integrating HIV sequence and drive distance data with traditional epidemiological information.

Applying this approach to the Botswana Combination Prevention Project (BCPP), the authors make the noteworthy finding that an estimated 90% [CI 81-93%] of new HIV infections that occurred in communities receiving combination prevention including test-and-treat originated from communities outside the trial area. This has far reaching implications for the design and interpretation of related trials as well as public health policy more broadly.

I have comments regarding the extent to which available data support the above estimate, as well as minor comments that relate to the data set and interpretation of findings, which I hope once addressed will improve what is a creative and compelling manuscript.

We thank the reviewer for their interest in this novel statistical approach that combines genetic epidemiology and classical epidemiological data to aid interpretation of mass public health interventions in cluster-randomized trials.

Major comments

1. The BCPP reported a more than 30% reduction in new HIV infections in the intervention group relative to the standard-care group (Makhema et al. 2019, DOI: 10.1056/NEJMoa1812281); how can this reduction be explained alongside the author's estimate here that 85.6% [CI 73.6 - 90.5%] of transmissions in the standard-care group arose from non-trial communities?

There would, assuming the above, be a subset of new HIV infections (14.4% [CI 9.5 - 26.4%]) for which a local intervention could theoretically have an impact. If the intervention was completely successful, the subset could reduce completely to give an overall reduction in new HIV infections of at most 14.4%; this does not reach the more than 30% reduction observed (Makhema et al. 2019) and makes the above estimate of 85.6% seem untenable.

We thank the reviewer for their comment. The reviewer is correct to point out the apparent 'discordance' between the randomized-controlled trial result of prevented infections (30%) and the modeled preventable transmissions by a local intervention (14.4%). One explanation is that the combination prevention intervention package included test-and-treat as well as voluntary

safe male circumcision (PMID: 32504575); however, the model assumes that the intervention mainly only prevents transmissions from intervention sources, that is, people with HIV in intervention communities. In practice, the intervention, through circumcision, averts also some portion of transmissions to recipients in intervention communities. To provide a like-for-like comparison we would need to unpack the individual contribution of each component in the combination prevention intervention package to the measured 30% reduction. To estimate the portion of the 30% reduction that is due to test-and-treat versus male circumcision versus other factors in the intervention. Because the BCPP study was designed to measure the effectiveness of the intervention package as a whole and not its constituent parts, such an analysis would be a substantial undertaking that is reserved for future work. In addition to this mechanistic point, we also note that both the RCT estimate and the estimate in the present analysis contain significant uncertainty, with considerable overlap in the confidence intervals of the maximum preventable infections 14% [9-26] and the estimate in the RCT of the proportion prevented 30% [10-54]. Note that the estimate of preventable infections is computed from the estimate that 86% [74 – 91] of transmissions in control communities occurred from non-trial communities as $14\% = 100\% - 86\%$. **To improve clarity, we have updated the limitations section in the discussion with the following statement:**

“... our model assumes that the intervention only prevents transmissions from intervention sources, that is, people with HIV in intervention communities. In practice, the intervention averts also some portion of transmissions to recipients in intervention communities through voluntary safe male circumcision [9]. Accounting for the impact of male circumcision could result in even more averted transmissions. In addition to this mechanistic point, we also note that both the RCT estimate and the estimate in the present analysis contain significant uncertainty, with considerable overlap in the confidence intervals of the maximum preventable infections 14% [9-26] and the estimate in the RCT of the proportion prevented 30% [10-54]. Note that the estimate of preventable infections is computed from the estimate that 86% [74 – 91] of transmissions in control communities occurred from non-trial communities as $14\% = 100\% - 86\%$.”

2. Secondly, the counterfactual modelling estimate that a nationwide application of the intervention could reduce transmission by 59% [CI 3-87%] has considerably wider confidence intervals than one could reasonably expect; as an example, given that combination prevention including test-and-treat has been shown as able to reduce transmission by more than 30% for an intervention group, as described (Makhema et al. 2019), there should be a strong prior that the intervention would have the same or greater impact when implemented nationally ($\geq 30\%$), due to transmission from neighbouring communities being reduced relative to localised interventions.

Can the authors explain how a nationwide application of the same intervention at the time of the BCPP could reasonably reduce HIV transmission by as little as 3%, as reflected in the above confidence intervals?

We value the reviewer's comment. One could reasonably assume (putting aside the wide confidence bounds around the estimate) that the measured 30% reduction in BCPP should provide a lower bound for transmissions that were preventable with a nationwide intervention. Indeed, such a lower bound would be for the combined contribution of both test-and-treat and male circumcision; however, the lower bound for either test-and-treat or male circumcision alone would likely be lower than 30%. The number of transmission pairs included in the model, together with the fact that the model assumes that the intervention only prevents transmissions from intervention sources, is consistent with wider confidence bounds. Given these complexities we did not attempt a Bayesian analysis to combine the RCT findings with the findings using the alternative approach in this study. Because we share what we take to be the reviewer's point about tighter confidence bounds potentially providing greater insight about the quantity of interest, a future study could augment the number of transmission pairs included in the model with direct transmission events between same-sex pairs. Out of the 153 highly supported probable transmission pairs identified in BCPP, there were 82 directed opposite-sex pairs and 71 same-sex pairs (65 female-female, 6 male-male) (PMID: 35229714). We restricted our analysis to opposite-sex pairs because of insufficient information to identify the number and sex of unsampled intermediates in same-sex pairs. **To improve clarity, we have added the following statements to the discussion:**

"To augment sample size and account for the impact of same-sex transmission in Botswana, a future study could include directed same-sex transmission pairs in the model of the risk of transmission between communities. Out of the 153 highly supported probable transmission pairs identified in BCPP, there were 82 directed opposite-sex pairs and 71 same-sex pairs (65 female-female, 6 male-male) [1]. We restricted our analysis to opposite-sex pairs because of insufficient information to identify the number and sex of unsampled intermediates in same-sex pairs at the time of analysis and reserve the inclusion of same-sex pairs for future study [1, 2]. Our model could be improved further by accounting for other HIV prevention interventions such as male circumcision, condoms and pre-exposure prophylaxis. That said our parsimonious model provides helpful insight into the impact of the BCPP intervention in reducing transmissions."

3. Further, if there are 'limited cross border introductions into Botswana', could the authors comment on why a nationwide application of combination prevention including test-and-treat could not reduce new HIV transmissions to a greater extent than estimated?

We thank the reviewer for their comment. Accounting for the impact of both test-and-treat and male circumcision in the model would likely result in even higher estimates of averted transmissions compared to accounting mainly only for the impact of test-and-treat as the model currently does. **To improve clarity, we have updated the discussion with the following statement:**

"Our model could be improved further by accounting for other HIV prevention interventions such as male circumcision, condoms and pre-exposure prophylaxis. That said our parsimonious model provides helpful insight into the impact of the BCPP intervention in reducing

transmissions.”

Minor comments

4. An introductory sentence detailing the available treatments and the standard of care during the study period would be beneficial, particularly as readers return to this manuscript in years to come.

We thank the reviewer for their suggestion and have **updated the “BCPP study description” section in the materials and methods with the following statement:**

“... control communities received the standard-of-care, which before 2016 meant that people with HIV qualified to start antiretroviral treatment when their CD4 cell count was below 350 cells per microliter. Beginning June 2016, the national HIV treatment policy was changed to universal treatment meaning that immediate antiretroviral treatment was now available in both arms of the BCPP trial. The first-line regimen, provided by the Government of Botswana to all trial communities, also changed from efavirenz (EFV)-tenofovir disoproxil fumerate (TDF)-emtricitabine (FTC) to dolutegravir (DTG)-TDF-FTC [1].”

5. The rates of directed transmission appear lower than one could expect; the authors report that 82 directed opposite-sex transmission pairs were identified, representing 2.1% (82/3832) of included sequences. Could the authors comment on this, potentially referencing previous work where relevant parameters have been identified or approaches calibrated?

We apologize for the insufficient clarity in our description and would like the reviewer to kindly note that out of the 3,832 trial participants that met minimum criteria for inclusion in phylogenetic analysis we identified 236 clusters of trial participants with genetically similar HIV-1 infections (525 trial participants / 3,832). We defined genetic similarity clusters as groups of two or more trial participants whose HIV-1 consensus viral whole genomes were separated by a distance at or less than 4.5% substitutions per site. The threshold of 4.5% substitutions per site was motivated by the distribution of genetic distances separating HIV-1 subtype C consensus viral whole genomes of epidemiologically linked couples in the HIV Prevention Trials Network (HPTN) 052 study (Please see section “Consensus sequence phylogenetics to identify clusters of participants with genetically similar HIV-1 infections” in Magosi and colleagues (PMID: 35229714)). From the mapped deep-sequencing short reads of the 525 trial participants within genetic similarity clusters we identified 153 highly supported probable transmission pairs (82 opposite-sex pairs, 71 same-sex pairs (65 female-female, 6 male-male)) and the probable direction of transmission between them. We defined transmission pairs with strong support for phylogenetic linkage and direction of transmission, that is highly supported pairs, based on a phylogenetic linkage and direction of transmission score threshold of 57%. The phylogenetic linkage and direction of transmission score threshold was informed by standard best practice such that the posterior probability for at least half of the windows along the genome supporting an ancestral relationship type between a pair of individuals, i and j exceeds 80% (please see section “Identifying probable source-recipient pairs with strong phylogenetic

evidence for linkage and direction of transmission” in methods of Magosi and colleagues (PMID: 35229714) and section “Classification of linked pairs and sources” in methods of Ratmann and colleagues (PMID: 30926780). **To improve clarity, we have updated the methods and results as follows:**

Methods:

“Deep-sequence phylogenetics data. ... The HIV-1 viral consensus whole genomes of individuals that met minimum criteria for inclusion in phylogenetic analyses were ones that had fewer than 30% of bases missing beyond the first 1,000 nucleotides ($\geq 6,300$ nucleotides available) (see “criteria for inclusion in phylogenetic analyses in [10]”). To efficiently use computational resources, viral consensus whole genomes were used to identify groups (or clusters) of trial participants with genetically similar HIV-1 infections as a filtering step to exclude distantly related sequences from deep-sequence phylogenetic analysis [10]. We defined clusters of genetically similar HIV-1 infections as groups of two or more trial participants whose viral whole-genome consensus sequences were separated by a genetic distance less than 4.5% nucleotide substitutions per site. This empirical threshold of 4.5% substitutions per site was motivated by the distribution of genetic distances separating subtype C HIV-1 viral whole genome consensus sequences of epidemiologically linked couples in the HIV Prevention Trials Network (HPTN) 052 study [11, 12] (see “Consensus sequence phylogenetics to identify clusters of participants with genetically similar HIV-1 infections in [10]”). A detailed description of the deep-sequence phylogenetic analysis is published in [10]. Briefly, we performed parsimony-based ancestral host-state reconstruction with the phyloscanner software [13, 14] to identify pairs of trial participants with genetically similar HIV-1 infections and the probable direction of transmission between them. Based on empirical data we defined transmission pairs with strong support for phylogenetic linkage and direction of transmission, that is highly supported pairs, based on a phylogenetic linkage and direction of transmission score threshold of 57% (see section “Identifying probable source-recipient pairs with strong phylogenetic evidence for linkage and direction of transmission” in [10]). We identified both highly supported same-sex and opposite-sex (female-to-male or male-to-female) transmission pairs, however, we restricted our analyses to opposite-sex pairs because we could not reliably distinguish between direct transmission in same-sex pairs, and same-sex members of transmission chains with unsampled intermediates [10]. For brevity, we refer to the directed opposite-sex transmission pairs as source-recipient pairs.”

Results:

“Of the 5,114 trial participants who consented to a blood draw for viral genotyping and whose HIV viral whole genomes were successfully deep-sequenced [1, 10], 3,832 met inclusion criteria for phylogenetic analysis, and from those, we identified 236 clusters of trial participants with genetically similar HIV-1 infections (525 / 3,832 trial participants). Within the 236 genetic similarity clusters we identified 82 directed opposite-sex transmission pairs between ordered pairs of the 30 communities in the BCPP trial (Supplementary Figure 1), we also identified 71 same-sex pairs between women ($n = 65$) and men ($n = 6$) [10]. Because the transmission of HIV

in Botswana and Southern Africa is predominantly through heterosexual contact, and direct transmission is rare between women, we assumed that same-sex pairs represent transmission chains with one or more unsampled intermediates [9, 10, 18]. Therefore, we restricted subsequent analyses to the directed opposite-sex transmission pairs. Of the 82 source-recipient pairs, 51 (21 female-to-male, 30 male-to-female) were identified between HIV viral genomes sampled during the baseline period of the trial compared to 31 (16 female-to-male, 15 male-to-female) where the recipient's genome was sampled post-baseline. We defined the post-baseline period as at least one year after baseline household survey activities had concluded in a community such that the intervention could have taken effect."

6. A reader may assume that same-sex transmission is negligible in Botswana on reading the manuscript; transmission pairs are only classified as heterosexual, with directionality as male-female or female-male, which contributes to the source attribution model and many findings from the manuscript. I do not believe that this was intended by the authors, although I believe that careful wording is required both in the manuscript methods and discussion to not reinforce cultural norms that may stigmatise and erase same-sex relationships in country; the authors could reference their own work in this area (Magosi et al. 2022), including the challenges of appropriately describing the inferred transmission between women in southern Africa, as well as other relevant works (Matlapeng et al. 2023, PMID: 37830348) to address this.

Kindly refer to the response to reviewer comment number 5. Furthermore, we have updated the discussion with the following statement:

"To augment sample size and account for the impact of same-sex transmission in Botswana, a future study could include directed same-sex transmission pairs in the model of the risk of transmission between communities. Out of the 153 highly supported probable transmission pairs identified in BCPP, there were 82 directed opposite-sex pairs and 71 same-sex pairs (65 female-female, 6 male-male) [10]. We restricted our analysis to opposite-sex pairs because of insufficient information to identify the number and sex of unsampled intermediates in same-sex pairs at the time of analysis and reserve the inclusion of same-sex pairs for future study [10, 22]."

7. Confidence intervals could be a worthwhile addition to Figure 4, given that the nationwide intervention estimates have a wide confidence overall.

We thank the reviewer for their suggestion. Because the estimated relative benefit of a nationwide roll-out of the BCPP intervention is consistently a 59% reduction in transmissions to recipients in trial communities from communities nationwide. **We thought it would be helpful to the reader to replace Figure 4 with the following text in the results:**

"Impact of a nationwide intervention

From the post-baseline model in Table 1 we estimate that the relative benefit of a national roll-out of the BCPP intervention would be a 59% [3 – 87] reduction in transmissions to recipients in

trial communities from communities nationwide. For example, we compute the relative benefit of the BCPP intervention from the post-baseline model in Table 1 as ($59\% = 1 - \exp(-0.9)$). Then 95% confidence intervals for the relative benefit can be computed in two ways. First, we used an empirical bootstrap approach to compute 95% confidence intervals from the 2.5% and 97.5% quantiles of the distribution of 1,000 bootstrap samples, 59% [3 – 87]. Second classical 95% confidence intervals were computed from Table 1 as (lower bound: $1 - \exp(-(-0.11))$), upper bound: $1 - \exp(-(1.90))$, 59% [-11 – 85]). Note that the difference in confidence interval estimates provided by the two approaches reflects the sample size of the data, a larger sample size would result in greater agreement between the two approaches. This finding adds evidence that the impact of the BCPP trial intervention could be substantially larger than that observed in the trial if applied nationally.”

8. There may need to be a purple confidence interval for the curve in Figure 1A.

We thank the reviewer for their suggestion and **we have increased the color intensity of the confidence bands for both intervention community sources (purple) and control community sources (green).**

9. The authors could comment on the practicalities of implementing a nationwide introduction, as well as the estimated benefits if achieved; are there limitations on staffing or resources that make localised introductions for feasible, and what would key measures of success be for the intervention given co-occurring changes to the standard care.

We thank the reviewer for their comment and happy to provide additional clarity. Interventions and strategies for HIV prevention and treatment in Botswana are delivered by the Ministry of Health through a highly decentralized system of health care facilities that are overseen by District Health Management Teams (DHMTs) of health care workers. This allowed the BCPP intervention to be deployed through existing infrastructure and health care personnel and would allow for a nationwide intervention to be administered the same way (PMIDs: 32504575, 29314658, 32091179). An initial nationwide intervention to maximize capture of sources of transmission could be done once followed by regular targeted interventions informed by genomic surveillance. Indeed, Botswana already conducts genomic surveillance as part of HIV drug resistance monitoring. Strengthened linkage-to-care through scheduling of expedited appointments, text alerts before appointments and follow-ups for missed appointments was a key component of the BCPP intervention (PMIDs: 32504575, 29314658, 32091179) that would be important for a nationwide intervention and subsequent targeted interventions. **To improve clarity, we have updated the discussion as follows:**

“Our findings have implications for public health policy and for the design of effective HIV prevention strategies.

... As highlighted in the BCPP study description in the methods, the BCPP intervention was deployed through a highly decentralized system of health care facilities that are overseen by District Health Management Teams (DHMTs) of health care workers in the Ministry of Health, therefore, a nationwide roll-out of the BCPP intervention could be administered in the same

way [9, 23, 24]. An initial nationwide intervention to maximize capture of sources of transmission could be done once followed by regular targeted interventions informed by genomic surveillance.”

10. The finding that partners in the same community contributed more to transmission in rural communities that are geographically isolated (Lines 240-242) seems guaranteed by the model, given the use of pairwise travel distance data. Is there a hypothetical scenario where this would not be the case?

We thank the reviewer for their comment and agree with the reviewer that the model guarantees this result (at least for communities of a given size).

Dr George Tairaoa

Reviewer #1 (Remarks on code availability):

The code seems readily available, although I will not have time to replicate the analyses here.

Reviewer #2 (Remarks to the Author):

This study provides a unique approach to evaluate the effect of population-level HIV prevention interventions on HIV transmission and a modelling approach to estimate potential benefits of intervention scale-up to a national level. The approach combines phylogenetic inference and statistical modeling. The findings have substantial public health importance and highlight the advantage of broader nation-wide interventions over localized ones. While the paper is very compelling, a few important limitations need to be addressed before publication.

We thank the reviewer for their interest in this novel statistical approach to assess public health interventions deployed at scale.

Major comments:

11. Given that the number of non-trial communities is much higher than trial communities (458 vs 30), it is expected by chance that those communities will contribute the majority of infections coming from the outside. I suggest permutation analysis with random assignment of “trial” status to communities, to allow generation of baseline expectation of contributions from non-trial communities.

We thank the reviewer for their suggestion. A permutation analysis is an interesting idea but is unfortunately infeasible. Permutation analysis depends on **randomization** and **exchangeability**, however, in this case **we do not have exchangeability** because the longitudinal HIV incidence cohort in control communities was tested annually to identify new cases of people with HIV

compared to non-trial communities that did not receive annual testing. Also, non-trial communities would need to be matched by size of community, age structure, access to health facilities and proximity to urban centers. The point of our highlighting this imbalance was not to imply something was special about trial communities but only to note that they are few compared to the total number (as the reviewer points out) and thus that the existence of cross-community transmissions involving trial communities (which can be detected by our method) is an indication of even more cross-community transmissions involving non-trial communities.

12. The modelling study assumes that all transmissions happen within the country and disregard foreign importations, which might have limited the estimated effect of a potential nationwide intervention.

We thank the reviewer for their comment. Cross-border external introductions are an important consideration that we kept in mind in our analysis. In the discussion section we write about phylogenetic clustering analysis conducted by the PANGEA-HIV consortium that showed limited cross-border external introductions into Botswana from neighboring countries. Such an analysis suggests that extending the BCPP trial intervention to all communities nationally to target more sources could effectively reduce the occurrence of new infections. **Please see an excerpt below from the discussion:**

“... a clustering analysis conducted by the PANGEA-HIV consortium on HIV-1 viral consensus sequences from the AHRI study population in South Africa, BCPP trial in Botswana, MRC study population in Uganda, PopART study population in Zambia and Rakai study population in Uganda found few clusters including cohorts from different countries. For example, only a single cluster comprising two members was identified between sequence samples from Botswana and Zambia [21]. The limited cross-border external introductions into Botswana suggest that extending the BCPP trial intervention to all communities nationally to target more sources could effectively reduce the occurrence of new infections.”

13. Why were only opposite-sex couples considered? Even in the case of underreporting of MSM contacts or drug use contacts, were same-sex pairs not discovered through the phylogenetic explorations? Would be great account for behavioral risk factors in the model.

We thank the reviewer for their comment. **Kindly refer to the response to reviewer comment numbers 5 and 6.** We value the reviewer’s suggestion and agree that accounting for behavioral risk factors in the model could be helpful. However, such information would be challenging to obtain nationally at the community level.

14. I don’t think that “transmitter’s” gender was taken into account, which assumes equal probability of infection for both and might not be a fair assumption. Considering both biological and epidemiological factors that might make infections from one gender to the other more likely might be important for population-level transmission inference.

The reviewer raises a good point. In a previous study we estimated the risk of transmission between trial communities by age and sex accounting for sampling variation (Please see figure 12 in [10]).

15. How do the authors account for the fact that proviral DNA may contain archived, non-infectious virus that does not reflect actively transmitted virus? Could using proviral DNA have introduced bias in estimated transmission links in the analysis?

We thank the reviewer for their comment and would like the reviewer to kindly note that the utility of proviral DNA for the analysis of HIV transmission dynamics and for the surveillance of transmitted drug resistance has been successfully demonstrated in multiple studies (PMID: 26041893).

Minor comments:

16. "Source of infection" when referring to individuals or populations can be problematic; suggest using non-stigmatizing language as "infections attributed to" or "possible lineage exporting location".

We thank the reviewer for their comment and would like the reviewer to kindly note that the terms "source" and "recipient" are standard terms in genomic epidemiology studies to understand the direction of spread of infections in populations. To accommodate the reviewer, we have made adjustments where practical in the paper to include additional descriptors such as **origin community** and **destination community**. **Please see an excerpt below from the methods:**

*"Pairwise drive distance data. Pairwise drive distances between ordered pairs of 488 communities in the 2011 Botswana population and housing census were successfully sourced from the google distance matrix application programming interface (API) with the mapsapi package v0.5.0 in R v4.1.1 [1]. Note that the 2011 census was the most recent census at the time of the trial. The 488 census communities included all 30 communities in the BCPP trial. Therefore, of the possible 488 x 488 ordered community pairs between the 488 census communities we sourced 488 x 30 ordered community pairs that had any of the 488 census communities as a source (**origin**) **community** and any of the 30 trial communities as a recipient (**destination**) **community**."*

17. Some of the results are overstated, as the observed reduction is within the CI of the modelled scenario (line 44-45)

We thank the reviewer for their comment. It is reassuring that the observed reduction of 30% is within the confidence bounds of the modeled relative benefit of a national roll-out of the BCPP intervention, 59% [3 – 87]. A future analysis with a larger size could provide tighter confidence bounds.

18. Was there a cutoff to determine “genetically similar viruses? (line 138)

We thank the reviewer for their comment. Yes, kindly refer to the response to reviewer comment number 5.

Reviewer #3 (Remarks to the Author):

We thank the reviewer for their time and effort.

Reviewer #4 (Remarks to the Author):

Unpacking sources of transmission in HIV prevention trials with deep-sequence pathogen data
BCPP/Ya Tsie study
NCOMMS-24-55987-T

Overall:

This study estimated the relative contribution of different sources of HIV infection in a community randomized trial of HIV prevention, conducted from 2013 to 2018, and concluded that 90% of new infections that occurred in individuals residing in intervention communities likely originated from individuals residing in communities outside of the trial area. As the authors note, the findings “provide insight on the extent to which the BCPP intervention was diluted by spillover infections from control communities and from communities outside the trial area.” This dilution effect adds to the challenge of contamination that occurs in implementation science studies when the control communities are also exposed to the intervention.

The related concepts of dilution and contamination in implementation science studies are well-known factors that have the potential to undermine the accurate assessment of an intervention’s impact [Boily MC et al., 2012]. In this analysis, however, the authors have taken these concepts one step further by using deep sequencing data and statistical analyses (especially of geographic distances) to quantify the role of dilution by spillover in the Ya Tsie study.

Since the Discussion section notes spillovers of far less than 90% in other studies, the paper implicitly acknowledges that the high level of spillover in Ya Tsie does not necessarily generalize to other settings at other points in time. The authors instead suggest that other prevention implementation science studies should also use similar deep sequencing and statistical methods to estimate the ‘true’ impact of a community intervention. If these methods require substantial financial resources, however, the concluding point may seldom be feasible.

Comments/Questions:

19. Matches in sequencing data: It's clear these were not perfect matches but how often did recipient sequences 'match' with only one source sequence? How often were there zero 'matches', or more than one 'match'?

We thank the reviewer for their comment and are happy to provide additional clarity. Kindly refer to the response to reviewer comment number 5 for a summary of the phylogenetic clustering analysis, and note that out of the 82 directed opposite-sex pairs inferred in the deep-sequenced sample, there was a single male individual who transmitted to two females that each reside in different communities from his own.

20. Selection bias:

a. Lines 124-130 describe which people living with HIV from the trial were invited to join the study by providing a sample for viral phylogenetic analysis but were any excluded? And of those invited, what percentage agreed?

We thank the reviewer for their comment and would like the reviewer to kindly note that all trial participants with HIV provided informed consent for HIV-1 viral genotyping.

b. The sample size in the end was large at 3,832 and all were from trial participants. About 25% of consenting participants had samples that were not successfully sequenced (3832/5114). Did the authors look at the demographic characteristics of those whose samples were sequenced versus (i) those who did not consent) and (ii) had consented but had samples that could not be sequenced? A comparison of the characteristics and how similar or different they are would help alleviate concern for selection bias.

We thank the reviewer for their comment and would like the reviewer to kindly note that the viral genomes of all 5,114 trial participants that provided a sample for viral genotyping were successfully deep-sequenced, and from those 5,114 trial participants, 3,832 met inclusion criteria for phylogenetic analysis. As indicated in the methods, trial participants included in phylogenetic analysis were ones that had sequences with fewer than 30% of bases missing beyond the first 1,000 nucleotides ($\geq 6,300$ nucleotides available) (see "criteria for inclusion in phylogenetic analyses" and "paired-end deep-sequencing of HIV viral genomes for phylogenetic analyses" in [10]). Including sequences with greater missingness could compromise phylogenetic inference.

21. Pre-/post-2016: It will strengthen the paper if the authors were stratified analyses by pre-/post- 2016, when test-and-treat became a national strategy. Did the contribution of transmission from non-trial sites become less pronounced post 2016? One would surmise that would be the case after widespread implementation of test-and-treat. If the authors can demonstrate this, their causal argument would be strengthened. There may be sample size issues for these stratified analyses but a sensitivity analysis on this would be helpful.

We thank the reviewer for their comment and are happy to provide additional clarity. Kindly note that 91% (1,240 / 1,367) of the trial participants that were sampled post-baseline were sampled after June 1, 2016, suggesting that the model presented in the paper provides insight about transmission patterns after the introduction of universal treatment beginning June 2016.

22. Urban/rural analysis: The authors indicated that in two rural communities, transmission within the same community vs. transmission across different communities was more common because they were isolated and far away from the three urban centers. Did this phenomenon occur in other communities? The authors did not present a formal urban/rural stratified analysis which could strengthen the causal claim.

We thank the reviewer for their comment. The reviewer raises a good point that could be better addressed in a future study with a larger sample size.

23. Key populations: There was no discussion on key populations, particularly men-to-men transmission. A mention of why this was not included would be helpful, given the outsized role key populations play in ongoing transmission.

We thank the reviewer for their comment. Kindly refer to the responses to reviewer comment numbers 5 and 6. Also, kindly note that most trial participants in the household survey reported that they had not engaged in transactional sex in the last 12 months, suggesting that the contribution of sex work to transmission patterns identified in the trial was limited.

Minor comments:

24. "Pairwise drive distance": it would be more intuitive to call this "Pairwise driving distance" and then explicitly define it in the Methods section (e.g., at approx. Line 148) as the estimated driving distance between each of 488 census communities and the 30 trial communities.

We thank the reviewer for their comment. As requested, we have updated the sub-section title in the methods "*Pairwise drive distance*" with the phrase with the phrase "***Pairwise driving distance data***".

25. One wonders if, instead of 2011 census data and 2013 AIDS indicator survey data, it would have made a difference in the results if the authors had been able to use data from time periods closer to the end date (2018) of the Ya-Tsie study.

We thank the reviewer for their comment and would like the reviewer to kindly note that the 2011 Botswana population and housing census and 2013 Botswana AIDS Impact Survey were the most recent surveys at the time. The next census took place in 2022 and the next survey was conducted in 2021.

26. Table 1: Rather than showing how coefficients are converted to % on page 12 in the text itself, the authors might consider including these calculations directly in Table 1 to help readers

with the flow of the text on page 12 and with the interpretation of Table 1.

We thank the reviewer for their suggestion and have updated Table 1 with the calculations as requested.

Reference:

Boily MC, Mâsse B, Alsallaq R, Padian NS, Eaton JW, Vesga JF, Hallett TB. HIV treatment as prevention: considerations in the design, conduct, and analysis of cluster randomized controlled trials of combination HIV prevention. *PLoS medicine*. 2012 Jul 10;9(7):e1001250.

Reviewer #4 (Remarks on code availability):

N/A

Reviewer #5 (Remarks to the Author):

Reviewer #5 (Remarks o

References

1. R Core Team, *R: A language and environment for statistical computing*. 2021, R Foundation for Statistical Computing: Vienna, Austria.

REVIEWERS' COMMENTS

Reviewer #1 (Remarks to the Author):

The authors have comprehensively and satisfactorily addressed my previous comments. I have no additional remarks, and commend them on the work.

Reviewer #1 (Remarks on code availability):

The code appears readily accessible although I do not have scope to recreate the analyses here.

Reviewer #2 (Remarks to the Author):

We appreciate the clarifications and additional explanations provided. The manuscript was improved following the recommendations from all reviewers, but we feel a few important points still need more attention. In particular, assumptions about how “transmitter’s” gender was handled in the current model, and the potential bias from using proviral DNA should be stated more openly as limitations so readers can better interpret the results. We outline these points below:

14. I don’t think that “transmitter’s” gender was taken into account, which assumes equal probability of infection for both and might not be a fair assumption. Considering both biological and epidemiological factors that might make infections from one gender to the other more likely might be important for population-level transmission inference.

Author’s response:

The reviewer raises a good point. In a previous study we estimated the risk of transmission between trial communities by age and sex accounting for sampling variation (Please see figure 12 in [10]).

Reviewer’s Comment:

Thank you for referring to your earlier study. However, the concern relates specifically to the current model that includes both trial and non-trial communities. It is still unclear if “transmitter’s” gender was incorporated into this model, or if male-female and female-male transmissions were modeled as equally likely.

We value the reviewer’s comment and are happy to provide additional clarification. Our model assumes a similar risk of transmission from male and female transmitters. In practice, there is likely to be heterogeneity in the risk of transmission by sex and age of the transmitter. Due to sample-size constraints our model would have been underpowered to delineate the risk of

transmission by the sex of the transmitter. Larger future studies could extend our modeling approach to account for the age and sex of the transmitter.

To improve clarity, we have updated the discussion with the following statement:

Our model assumes a similar risk of transmission from male and female transmitters. In practice, there is likely to be heterogeneity in the risk of transmission by sex and age of the transmitter. To broaden insights, our statistical modeling approach could be applied to estimate the relative contribution of various sources of infection by age and sex in the other community-randomized universal HIV test-and-treat trials that have assembled deep-sequence genomic data, for example the PopART trial in South Africa and Zambia.

15. How do the authors account for the fact that proviral DNA may contain archived, noninfectious virus that does not reflect actively transmitted virus? Could using proviral DNA have introduced bias in estimated transmission links in the analysis?

Author's response:

We thank the reviewer for their comment and would like the reviewer to kindly note that the utility of proviral DNA for the analysis of HIV transmission dynamics and for the surveillance of transmitted drug resistance has been successfully demonstrated in multiple studies (PMID: 26041893).

Reviewer's Comment:

Thank you for the provided reference. However, that study used proviral DNA sequences to determine phylogenetic clustering rather than directionality of infection, so it does not address the concern here. Would the timing of viral suppression bias the analysis of deep-sequencing data in phyloscanner? Could you cite any validation of proviral DNA vs. RNA (active) virus for the specific inference task (who infected whom)? Or discuss the potential bias from archived proviral DNA and acknowledge as a potential limitation in the manuscript.

We thank the reviewer for their comment. The reviewer raises a good point. We cannot rule out possible bias in the use of proviral DNA for the inference task of who infected whom, however, the successful use of proviral DNA in viral evolution studies suggests such bias could be limited. Future studies could systematically compare proviral DNA and viral RNA for the inference task of who infected whom.

To improve clarity, we have updated the discussion with the following statement:

We cannot rule out possible bias in the use of proviral DNA for the inference task of who infected whom, however, the successful use of proviral DNA in viral evolution studies suggests such bias could be limited. Future studies could systematically compare proviral DNA and viral RNA for the inference task of who infected whom.

We also believe that the response to Reviewer 4's comment number 20b is not satisfactory, as the same analysis (comparing demographic characteristics of those whose samples were

included in the analysis vs those who were not) could be helpful to demonstrate lack of bias in sample selection. This does not suggest “including sequences with greater missingness” as authors interpreted, but would allow better context of how included and excluded sequences differ.

We thank the reviewer for their comment and would like the reviewer to kindly note that the only samples we excluded were ones that had more than 30% of bases missing across the genome beyond the first 1,000 nucleotides or had missing sampling dates. There was little evidence to suggest a systematic difference in the demographic characteristics of sequence samples with missingness greater than 30% and sequence samples with missingness less than 30%.

Reviewer #3 (Remarks to the Author):

Reviewer #4 (Remarks to the Author):

The authors have responded to all of the reviewers' comments. I have no further comments for the authors.

Reviewer #5 (Remarks to the Author):
